# Glucosamine Use Is Associated with a Higher Risk of Cardiovascular Diseases in Patients with Osteoarthritis: Results from a Large Study in 685,778 Subjects

**DOI:** 10.3390/nu14183694

**Published:** 2022-09-07

**Authors:** Huan Yu, Junhui Wu, Hongbo Chen, Mengying Wang, Siyue Wang, Ruotong Yang, Siyan Zhan, Xueying Qin, Tao Wu, Yiqun Wu, Yonghua Hu

**Affiliations:** 1Department of Epidemiology and Biostatistics, School of Public Health, Peking University Health Science Center, Beijing 100191, China; 2School of Nursing, Peking University, No. 38 Xueyuan Road, Beijing 100191, China; 3Medical Informatics Center, Peking University, Beijing 100191, China

**Keywords:** glucosamine, cardiovascular disease, osteoarthritis, epidemiology

## Abstract

Glucosamine is widely used around the world and as a popular dietary supplement and treatment in patients with osteoarthritis in China; however, the real-world cardiovascular risk of glucosamine in long-term use is still unclear. A retrospective, population-based cohort study was performed, based on the Beijing Medical Claim Data for Employees from 1 January 2010 to 31 December 2017. Patients newly diagnosed with osteoarthritis were selected and divided into glucosamine users and non- glucosamine users. The glucosamine users group was further divided into adherent, partially adherent, and non-adherent groups according to the medication adherence. New-onset cardiovascular diseases (CVD) events, coronary heart diseases (CHD), and stroke, were identified during the observational period. COX proportional regression models were used to estimate the risks. Of the 685,778 patients newly diagnosed with osteoarthritis including 240,419 glucosamine users and 445,359 non-users, the mean age was 56.49 (SD: 14.45) years and 59.35% were females. During a median follow-up of 6.13 years, 64,600 new-onset CVD, 26,530 CHD, and 17,832 stroke events occurred. Glucosamine usage was significantly associated with CVD (HR: 1.10; 95% CI: 1.08–1.11) and CHD (HR: 1.12; 95% CI: 1.09–1.15), but not with stroke (HR: 1.03; 95% CI: 0.99–1.06). The highest CVD risk was shown in the adherent group (HR: 1.68; 95% CI: 1.59–1.78), followed by the partially adherent group (HR: 1.26, 95% CI: 1.22–1.30), and the non-adherent group (HR: 1.03; 95% CI: 1.02–1.05), with a significant dose–response relationship (*p*-trend < 0.001). In this longitudinal study, adherent usage of glucosamine was significantly associated with a higher risk for cardiovascular diseases in patients with osteoarthritis.

## 1. Introduction

Glucosamine is a nutritional supplement for joint cartilage, which is widely used for managing the symptoms of osteoarthritis [1,2] and is regularly consumed in approximately one-fifth of adults in the United States, Australia, and the United Kingdom [3,4,5]. Despite low potential toxicity reported in previous studies, the patients exposed to glucosamine experienced increased fasting blood glucose and reduced insulin sensitivity [6,7,8,9]. Biochemical studies have shown that glucosamine is an inhibitor of nitric oxide (NO) synthesis [10], which might affect microvascular remodeling and endothelial function regulation [11,12], and cause glucosamine a potential risk factor of cardiovascular disease. Since patients with osteoarthritis are at a high risk of cardiovascular disease [13,14,15], it is necessary to know whether glucosamine use in patients with osteoarthritis brings an additional risk for cardiovascular disease (CVD).

Previous randomized control trials focusing on evaluating the efficacy did not find excess cardiovascular risks of glucosamine compared with placebo or celecoxib in patients with osteoarthritis [16,17,18,19,20]. Nevertheless, these studies had a limited sample size and contained a follow-up period of no more than 2 years [16,17,18,19,20]. Recently, large cohort studies in the general population were designed to explore the association of habitual glucosamine use with the risk of CVD [5,21]. However, in these studies, the use of glucosamine was reported by participants without detailed records of the dosage and duration, and the association of glucosamine with cardiovascular events in patients with osteoarthritis is still unclear. Therefore, based on a comprehensive database with medication information from all hospitals, pharmacies, and medical facilities, we aimed to assess the association of glucosamine use with CVD in 685,778 patients with osteoarthritis in a real-world setting in Beijing, China.

## 2. Materials and Methods

### 2.1. Data Source

A retrospective cohort study was performed from 1 January 2010 to 31 December, based on the Beijing Medical Claim Data for Employees (BMCDE). The database was described elsewhere [22]. Briefly, it contained all of the medical and pharmacy records for about 20 million residents who enrolled in the urban employee basic medical insurance (UEBMI) program in Beijing. The UEBMI is basic medical insurance covering more than 92% of urban employees, workers, and retirees in China [23,24,25]. The BMCDE database includes sale information of drugs from all sources in Beijing, including all of the hospitals and retail pharmacies, and part of the data records from the database were manually validated by comparing with the original medical and pharmacy files. Demographic, diseases, and detailed medical information were derived from the database. All of the data were collected for administrative purposes without any personal identifiers. Therefore, this study was exempted from ethics committee review by the Ethics Committee of the Peking University Health Science Center.

### 2.2. Population

Of the 20.8 million residents enrolled in UEBMI, those who were newly diagnosed with osteoarthritis from 1 January 2010 to 31 December 2012, were selected. Then, the participants who newly started taking glucosamine within one year after diagnosis of osteoarthritis were considered as the exposure group (the distribution of period time from diagnosis of osteoarthritis to starting taking glucosamine is shown in Appendix A), and those who never use glucosamine were used as the control group. A two-year wash-out period was used to identify the new patients with osteoarthritis and the new glucosamine users, which meant that the participants diagnosed with osteoarthritis or taking glucosamine within two years before baseline were excluded as old cases or former users. Newly diagnosed osteoarthritis was determined by the International Classification of Diseases (ICD)-10 code (M0–M2). To control potential bias caused by the disease course and to estimate the long-term effect of glucosamine on CVD, those under 18 years or with <1-year follow-up time were excluded in both the glucosamine group and non-glucosamine group. Missing data were defined as missing any of the covariates, and the participants with missing data were excluded.

### 2.3. Exposure, Outcome, and Covariates

Of 685,778 participants from 774,912 patients newly diagnosed with osteoarthritis, there were 240,419 who newly started taking glucosamine within one year after diagnosis of osteoarthritis, and 445,359 patients never took glucosamine, while 89,134 patients who started taking glucosamine more than one year after the diagnosis of osteoarthritis were excluded from the main analyses (Figure A1). To further compare the differences among the groups with different adherence, the glucosamine group was further classified into adherent, partially adherent, and non-adherent groups. The proportion of the days covered (PDC) was used to quantify the adherence of glucosamine, based on the dosage of prescription recorded in the database and the daily defined dose (DDD) [26,27]. The PDC was calculated as the proportion of days on which a person had an available supply of glucosamine, which represents the adherence to which a person continued to fill glucosamine prescriptions over time. According to PDC, those who used glucosamine were divided into three subgroups, including adherent (PDC ≥ 80%), partially adherent (20% ≤ PDC < 80%), and non-adherent (PDC < 20%) [26].

The incident CVD, coronary heart disease (CHD), and stroke events of the subjects were observed from the index date to 31 December 2017, which were identified by the primary diagnosis at each inpatient or outpatient encounter during the follow-up. Each participant had only one outcome in this study. The primary outcomes were overall cardiovascular diseases, with an ICD code of I00-I99. The secondary outcomes were CHD and stroke, with an ICD code of I20-I25 or I60-I64, respectively.

Several covariates were adjusted, including age, sex, hypertension, medication, Charlson Comorbidity Index (CCI), and health care utilization index (HCUI) at the baseline [28]. Hypertension was defined as diagnosis with an ICD-10 code of I10-I15 or having anti-hypertensive prescriptions before the index date. Medications included lipid treatment, antiplatelet treatment (including aspirin, clopidogrel, indobufen, ticagrelor, ticlopidine, prasugrel, and cangrelor), and NSAIDs’ therapy (except aspirin and indobufen), which was determined using the pharmacy records before the index date. CCI is a weighted score categorizing and integrating comorbidities of patients based on the ICD diagnosis codes, including 19 comorbidities [29]. The higher scores indicated more comorbidities and poorer health status, and more details about CCI are shown in Appendix A. HCUI was calculated as the frequency of hospital visits in the past 12 months before the index date. Moreover, considering the indication bias of adherence, we selected the PDC of antidiabetic drugs as a negative control adjusted in the sensitivity analysis. The PDC of antidiabetic drugs was calculated in the same way as that mentioned before.

### 2.4. Statistical Analysis

The normal continuous variables were performed by the mean (standard deviation, SD), non-normal continuous variables by the median (interquartile range, IQR), while the categorical variables were represented by the number (percentage). The baseline characteristics were compared between groups using ANOVA, Wilcoxon test, or χ^2^ test. The index date was defined as the date of the first diagnosis of osteoarthritis in the non-glucosamine group, or the date when the patients first used glucosamine in the glucosamine group. The follow-up period was determined from the index date to the outcome occurrence or to 31 December 2017, which came first. The incidence rate of cardiovascular events was calculated and presented with 95% confidence intervals per 1000 person-years. The 95% CI of the incidence rate was calculated using the following formula:95% CI = the number of events/cumulative person-years ± 1.96 × sqrt (the number of events)/cumulative person-years(1)

Hazard ratios (HRs) and 95% CIs were reported to estimate the risk of cardiovascular events using the Cox proportional regression model after adjusting for age, sex, hypertension, medication, CCI, and HCUI. Sex (female or male), hypertension, lipid treatment, antiplatelet treatment, and NSAIDs’ therapy (yes or no) were analyzed as binary variables, while age, CCI, and HCUI were the continuous variables. Subgroup analysis stratified by age and sex in different adherence groups was then performed after adjusting for the other covariates.

Five sensitivity analyses were performed to prove our results. First, the 1:2 propensity score matching (PSM) within a caliper of 0.03 was conducted between the adherent glucosamine users and non-glucosamine users to reduce potential indication bias. The propensity scores for each participant between the groups were calculated using a logistic regression model including the index year and all of the covariates mentioned before. The patients were matched according to the propensity score, and the association was then reported in the matched patients. Second, to avoid the indication bias caused by adherence, the PDC of antidiabetic drugs was additionally adjusted as a negative control on the original model, to reduce the potential confounding. Third, patients with lower than 2 years follow-up time were excluded to further explore the long-term effect of glucosamine. Fourth, we included all of the patients newly diagnosed with osteoarthritis whether or not they took glucosamine within one year. Fifth, to validate our results and test potential residual confounding due to the course of the disease, follow-up time was defined separately as the period from the diagnosis of osteoarthritis to the outcomes.

A two-tailed *p* value less than 0.05 was considered statistically significant. PSM was analyzed by the “MatchIt” package in R 3.6.3 with a caliper of 0.03. Otherwise, the statistical analyses were completed using SAS version 9.4.

## 3. Results

### 3.1. Characteristics of Study Population

A total of 685,778 participants with a new diagnosis of osteoarthritis were included in the main analysis. The average age was 56.49 (SD: 14.45) years, 59.35% were female, 21.20% presented with hypertension, 37.29% were undergoing lipid treatment, 27.74% were undergoing antiplatelet treatment, 29.67% were undergoing NSAIDs therapy, the mean of the CCI was 0.67 (SD: 0.91), and the median of HCUI was 21 (IQR: 6, 45). Among them, 240,419 subjects were glucosamine users and 445,359 were non-users, respectively. Table 1 shows the characteristics of study subjects according to their glucosamine-using status. When compared with non-users, the glucosamine users were older, more likely to be females, more likely to have hypertension, more likely to have lipid treatment or antiplatelet treatment, less likely to have NSAIDs therapy, had higher CCI, and had lower HCUI.

### 3.2. Incidence of CVD Events

The median follow-up time was 6.13 (IQR: 5.57–6.75) years overall, 6.15 (IQR: 5.61–6.81) for non-users, and 6.01 (IQR: 5.48–6.68) for glucosamine users. There were 64,600 new-onset CVD events, with an overall incidence rate of 15.61 (95% CI: 15.49–15.73) per 1000 person-years. There were 26,530 and 17,832 CHD and stroke events, with incidence rates of 6.25 (95% CI: 6.18–6.33) and 4.18 (95% CI: 4.12–4.24) per 1000 person-years, respectively. Compared with the non-users, the glucosamine users had significantly higher incidence rates of overall CVD events, CHD, but not stroke (Table 2). Among the glucosamine users with different adherence levels, a higher incidence was observed in the group with a higher adherence (Appendix A).

### 3.3. Association of Glucosamine with Cardiovascular Events

The association of glucosamine with cardiovascular events is shown in Figure 1 and model adjustment is shown in Table A1. Glucosamine usage was significantly associated with overall CVD (HR: 1.10, 95% CI: 1.08–1.11), CHD (HR: 1.12, 95% CI: 1.09–1.15) but not stroke (HR: 1.03, 95% CI: 0.99–1.06). The cardiovascular risks for glucosamine use with different adherence levels were further explored. The highest HR was observed in the adherent group for CVD (HR: 1.68, 95% CI: 1.59–1.78), CHD (HR: 1.69, 95% CI: 1.56–1.84), and stroke (HR: 1.53, 95% CI: 1.37–1.70) followed by the partially adherent group, then the non-adherent group, compared to non-users. A significant dose–response relationship was shown for glucosamine with CVD, CHD, and stroke events in all of the subjects (*p*-trend < 0.001 for each).

The subgroup analysis for CVD is shown in Appendix A. The similar trends among the glucosamine users with different adherence were found when stratified by sex or age (*p* for trend < 0.001 for each).

### 3.4. Sensitivity Analysis

After the 1:2 PSM, 8022 adherent glucosamine users and 16,043 matched glucosamine non-users were included, while one glucosamine user was matched to only one control due to the strict caliper. The baseline characteristics showed no significant difference (Appendix A), and the adherent glucosamine use was still significantly associated with CVD, CHD, and stroke (Appendix A). The results were consistent with those in the main analysis when the PDC of antidiabetic drugs was additionally adjusted considering the indication bias (Appendix A), or when patients with lower than 2 years follow-up time were excluded (Appendix A). Among 774,912 patients newly diagnosed with osteoarthritis whether or not they took glucosamine within one year, the glucosamine use was significantly associated with CVD, CHD, and stroke with a significant trend for each (*p* for trend < 0.001) (Appendix A). Lastly, to avoid potential confounding due to the course of disease, follow-up time was defined separately as the period from the diagnosis of osteoarthritis to the outcomes, and a significant association was also estimated for CVD, CHD, and stroke (Appendix A)

## 4. Discussion

Despite several studies [5,21] that were conducted to evaluate the association of glucosamine with CVD risks in the general population, the association was still unclear in patients with osteoarthritis who were at a higher risk of cardiovascular events. In this retrospective cohort study, based on a comprehensive database with prescription information for nearly 0.7 million patients newly diagnosed with osteoarthritis, we assessed CVD, CHD, and stroke risks between glucosamine users and non-users in patients with osteoarthritis. According to the results, we found that glucosamine was significantly associated with a higher risk for CVD and CHD, especially in patients who had a higher adherence. Although no statistically significant association of glucosamine use with stroke was found, a 53% increase in the risk of stroke was estimated in adherent glucosamine users significantly. Taking glucosamine was already strongly recommended against in patients with osteoarthritis in America [1], and was given a weak recommendation in Europe [2] by Guidelines, due to its limited efficacy. The present study provides additional safety considerations for long-term glucosamine use for the guidelines.

Most importantly, this study provides new references for screening people at high risk of cardiovascular diseases in patients with osteoarthritis. The association of glucosamine usage and CVD risks among patients with osteoarthritis in previous studies was inconsistent. Several randomized clinical trials reported no significant association between glucosamine and CVD risks in patients with osteoarthritis [16,17,18,19,20], but the evidence was insufficient due to the lack of sample size and the short follow-up of no more than 2 years. The randomized clinical trials with a three-year follow-up found that there was no significant difference in the levels of blood pressure, lipids, and glucose between patients with osteoarthritis taking crystalline glucosamine sulfate and those with placebo [30], but these biochemical markers are not synonymous with cardiovascular events. Recently, two cohort studies reported habitual glucosamine users had a lower risk for CVD in general population [5,21]. However, patients with osteoarthritis were at higher risk for cardiovascular events [13,14,15], which had potentially mechanistic differences from general population. Meanwhile, another cohort study using the same database as the above two studies found that glucosamine use was associated with a lower incidence of type 2 diabetes in patients who initially were free of diabetes, cancer, or cardiovascular disease at baseline. However, again, these participants were rather healthy at baseline [31]. The results of those studies could not be directly extrapolated to those in patients with osteoarthritis, which might cause the main difference from our research. In the present research, the association was estimated by the patient-based design and the adjustment for comorbidities, medications, and health-care utilization. Moreover, a dose–response relationship between glucosamine and cardiovascular events was first found in this study, indicating that glucosamine use and its adherence are important when considering the risk for CVD in patients with osteoarthritis. Previous studies have not analyzed the association of glucosamine with cardiovascular events in terms of medication adherence. In this research, analyzing the different adherence groups separately rather than together helped to ensure the authenticity of the results. Although the relationship should be interpreted with caution due to the difference of characteristics among the groups, we performed multiple adjustments of covariates and subgroup/sensitivity analyses to minimize the differences, and our results were stable in all of the analyses. Further research is still needed to determine the association. For specific cardiovascular disease, different effects of glucosamine were observed between CHD and stroke, which might indicate different biological and other aspects of different events.

A higher CVD risk in glucosamine users among the patients with osteoarthritis is also biologically plausible. First, previous studies showed that glucosamine may increase fasting blood glucose, accelerate atherosclerosis, and reduce insulin sensitivity [6,7,8,9,32,33,34,35,36,37], which have been widely identified as risk factors for cardiovascular diseases [38,39,40,41]. Second, glucosamine could inhibit the synthesis of nitric oxide (NO) [10]. As a protective signaling molecule, NO plays an important role in preventing atherosclerosis [42]. Suppression of NO might accelerate atherosclerosis.

This large population-based cohort study was based on comprehensive medical and pharmacy information for nearly 20 million residents in Beijing, China. During a median observational period of 6.13 years, the long-term CVD risks were assessed in patients using glucosamine with different adherence levels. Due to the comprehensive prescription-recording system, covering all of the pharmacy records from all of the hospitals and pharmacies, the glucosamine-taking frequency can be obtained in detail. Additionally, multiple sensitivity analyses were performed to validate the results. Although the huge number of participants showed a relatively low difference in outcome between groups with low *p*-values, we still found adherent glucosamine users had a 68% increase in risk for CVD compared with that in non-users. The results provide real-world evidence on the association of glucosamine use with CVD events and insights into screening individuals with a high CVD risk in patients with osteoarthritis based on a medical insurance database.

This study has several limitations. First, the participants in this research were all from Beijing, the capital city of China, who had a relatively high income and convenient medical services. The interpretation of the results should be cautious. Second, the adherence levels of glucosamine users were determined by the prescription dispensation. The dispensation did not necessarily indicate the actual drug consumption. However, any underestimation was likely to have been the same across groups, meaning that our results were conservative. Meanwhile, the dispensations outside the system could not be obtained, but the possibility was low. Third, the severity of osteoarthritis, BMI, physical activity, alcohol, and smoke status, etc. could not be obtained in our database, which was an inherent drawback of insurance database research and caused potential confounding. We could not rule out whether glucosamine users, especially those who used adherently, had more severe osteoarthritis, less physical activity, or a higher BMI, i.e., they were at a higher risk of CVD. However, in the sensitivity analysis of this study, our results were stable after matching covariates including hypertension, CCI, and HCUI, which were closely correlated with osteoarthritis severity, physical activity, and BMI at baseline. The relativity was helpful to reduce the potential confounding. Although some previous studies considered that the PSM was not the best matching method [43], our results still showed a high degree of consistency after matching. Meanwhile, in another sensitivity analysis, we adjusted PDC for antidiabetic drugs to reduce the potential bias due to the patients’ health status and severity of osteoarthritis. Moreover, the genetic aspects affecting risk of CVD could not obtained in our database, but we considered them to be a nondifferential misclassification between the groups, which made the results more conservative. Further research is still needed to prove our results with more covariates. Furthermore, the CVD risk of the combination of glucosamine with other medication, such as chondroitin, is still needed to be explored in the future.

## 5. Conclusions

In this longitudinal study, adherent usage of glucosamine was significantly associated with a higher risk for CVD in patients with osteoarthritis. The results suggested that the risks and benefits of glucosamine need to be revisited. Considering the potential residual confounding, the findings should be interpreted cautiously.

## Figures and Tables

**Figure 1 nutrients-14-03694-f001:**
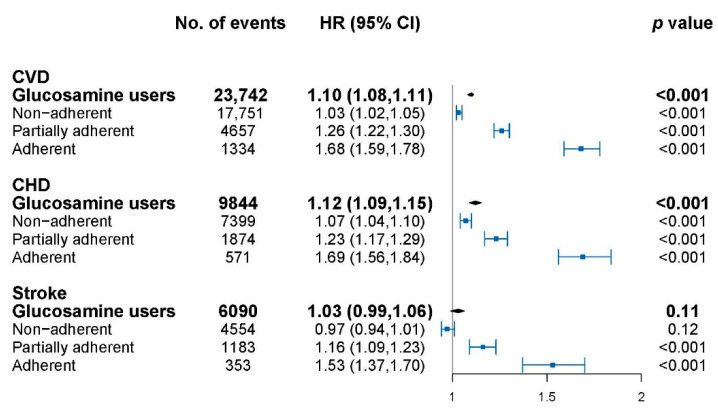
Association of glucosamine users with cardiovascular disease (CVD) at different adherence levels. CVD, cardiovascular disease; CHD, coronary heart disease; HR, hazard ratios; CI, confidence interval. HRs were calculated compared with glucosamine non-users, and were adjusted for age, sex, hypertension, medication, Charlson Comorbidity Index (CCI), and health care utilization index (HCUI). The dots indicate the HRs and the lines between bars represents the 95% confidence intervals.

**Table 1 nutrients-14-03694-t001:** Characteristics of the study population.

Characteristics	Glucosamine Non-Users (*n* = 445,359)	Glucosamine Users (*n*= 240,419)	*p* Value
Age, years, mean (SD)	55.44 (15.40)	58.43 (12.24)	<0.001
Female, *n*(%)	247,181 (55.50)	159,826 (66.48)	<0.001
Hypertension, *n*(%)	89,394 (20.07)	55,974 (23.28)	<0.001
Lipid treatment, *n*(%)	153,169 (34.39)	102,571 (42.66)	<0.001
Antiplatelet treatment ^1^, *n*(%)	113,118 (25.40)	77,084 (32.06)	<0.001
NSAIDs therapy ^2^, *n*(%)	140,099 (31.46)	63,387 (26.37)	<0.001
CCI, mean (SD)	0.65 (0.93)	0.71 (0.86)	<0.001
HCUI, median (IQR)	22 (7, 45)	20 (6, 45)	<0.001

NSAIDs, non-steroidal anti-inflammatory drug; CCI, Charlson Comorbidity Index; HCUI, health care utilization index; IQR, interquartile range. ^1^ Antiplatelet treatment included usage of aspirin, clopidogrel, indobufen, ticagrelor, ticlopidine, prasugrel, and cangrelor. ^2^ NSAIDs therapy here did not include aspirin and indobufen.

**Table 2 nutrients-14-03694-t002:** Incidence of cardiovascular disease events.

	Glucosamine Non-Users		Glucosamine Users		*p* Value ^2^
	Number of Events	Incidence ^1^ (95% CI)	Number of Events	Incidence ^1^ (95% CI)	
Overall CVD	40,858	15.08 (14.93, 15.23)	23,742	16.63 (16.42, 16.84)	<0.001
CHD	16,686	6.01 (5.91, 6.10)	9844	6.72 (6.59, 6.85)	<0.001
Stroke	11,742	4.21 (4.13, 4.28)	6090	4.13 (4.03, 4.23)	0.57

CVD, cardiovascular disease; CHD, coronary heart disease; CI, confidence interval. ^1^ “Incidence” indicates crude incidence without adjusting for any covariates. ^2^ “*p* values” were calculated for the difference between incidences of two groups.

## Data Availability

Not applicable.

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
