# Peer review of "Glucosamine Use Is Associated with a Higher Risk of Cardiovascular Diseases in Patients with Osteoarthritis: Results from a Large Study in 685,778 Subjects"

_nutrients, 2022, doi:10.3390/nu14183694_

Round 1

Reviewer 1 Report

The present study is touching an important topic in the biomedical food science, being the potentially unwanted effects of consuming health improving ingredients. The authors use a Chinese database containing parameters of a huge number of persons. This represents also a risk: with these numbers a relatively low difference in outcome between groups will automatically lead to extremely low p-values. Although the manuscript is very readable this reviewer would appreciate some editing in the text.

Major issues:

-          My main concern of the present study is the impact by confounding parameters on the outcome, especially in the light of the large number of persons involved. This reviewer would suggest to model the consumption of glucosamine with the various potentially available confounding factors instead applying stratification. To which extent is the consumption of glucosamine significantly associated with the dependent parameter in the presence of other significantly explaining factors. E.g. is age or genetic burden or any other factor not stronger related with the outcome? Therefore a model fit with all available significant parameters could strengthen the message when glucosamine remains significantly despite the presence of other significant factors. The fit of the model would be an important factor to explain the incidence of CVD. Why not use this approach?

-          As the authors know: the extremely high numbers of participants result in extremely low p-values (Table 1), therefore a thorough evaluation of the impact by confounding factors within one overall model would be strongly encouraged.

-          The present reviewer appreciates the mentioning of the specific patient group of persons with osteoarthritis in many sections of the manuscript by the authors. The outcome of the present study only applies on this specific group. As presented in the Discussion section the population consists of somewhat older adults suffering from osteoarthritis. Unfortunately no physical burden of this disease is mentioned but it could well be that the pain per participant is very different which could result in the use of all kinds of treatments in those persons with the heaviest burden and having its impact on the outcome. In Table 1 it is clear that the those persons consuming glucosamine are the ones with the highest comorbidities. The question then becomes to which extent this aspect is covered by the covariates, such as CCI and HCUI. Please provide more thoughts on the impact of characteristics of the participants on the outcome.

-          Do the authors believe that important covariates were not present in the database, such as information on genetic aspects of acquiring CVD issues?

-          In Europe the combination of glucosamine with chondroitin is very often used. Could the authors comment to which extent their findings would be different between persons with and without chondroitin or is there no information in the database on this parameter?

-          In the present manuscript we are dealing with an incidence parameter and not with a prevalence parameter: the authors were looking for new cases. Would the authors believe that there might be a difference between incidence versus prevalence scoring with respect to the consumption of glucosamine on the risk of acquiring CVD issues?

-          Please provide some explanation with respect to Table 2, more specifically how is the Confidence Interval calculated? Moreover is it possible that within one person more than one incidence score (new cases after treatment of older cases) could be detected? What is the definition here.

-          In that light please explain lines 80 – 81 more thoroughly: what is specifically meant here?

-          Instead of the construction of cohorts (various adherence groups) for the glucosamine adherence group why not use the adherence scores as a continuous factor? The power of this approach might be substantially higher than the one selected by the authors.

-          Another aspect in the above is the application of the propensity score matching. There are however authors who suggest that this PSM is not the best way to counteract selection bias and causal estimation, such as King&Nielsen (2019), or omitted variable bias. This reviewer appreciates some more discussion on this topic, such as in the Discussion section.

-          Sensitivity analyses as performed. Please discuss to which extent the high number of persons/person years limit the applicability of these analyses.

Minor issues:

-          Please erase lines 154 – 156

-          Grammar issues, e.g.

o   Line 18: “were selected out” versus “were selected”

o   Line 44: “were” versus “are”

o   Lines 49 – 50: “were with” versus “contained”

o   Lines 63: “it collected” versus “it contained”

o   Line 91: “take” versus “took”

o   Line 205: “was showed” versus “showed”

o   Etc.

Author Response

Dear Prof. Bruno Trimarco, Prof. Dr. Gaetano Santulli,

Thank you so much for your review concerning our manuscript entitled “Glucosamine use and risk of cardiovascular diseases in patients with osteoarthritis: a longitudinal study in China”. We gratefully thank all editors and reviewers for your time and useful suggestions. The comments are very precious, valuable, and helpful, which would significantly improve the quality of our manuscript. In addition, one reviewer suggested that our manuscript needs extensive English revisions, and we accept any form of English revisions. We have read the comments carefully and uploaded files of the revised manuscript. Revisions in the text are highlighted in ‘track changes’ and the responses to the comments are as follows.

Response to reviewers

#Reviewer 1

  • Comments

The present study is touching an important topic in the biomedical food science, being the potentially unwanted effects of consuming health improving ingredients. The authors use a Chinese database containing parameters of a huge number of persons. This represents also a risk: with these numbers a relatively low difference in outcome between groups will automatically lead to extremely low p-values. Although the manuscript is very readable this reviewer would appreciate some editing in the text.

Major issues:

-      Comment 1: My main concern of the present study is the impact by confounding parameters on the outcome, especially in the light of the large number of persons involved. This reviewer would suggest to model the consumption of glucosamine with the various potentially available confounding factors instead applying stratification. To which extent is the consumption of glucosamine significantly associated with the dependent parameter in the presence of other significantly explaining factors. E.g. is age or genetic burden or any other factor not stronger related with the outcome? Therefore a model fit with all available significant parameters could strengthen the message when glucosamine remains significantly despite the presence of other significant factors. The fit of the model would be an important factor to explain the incidence of CVD. Why not use this approach?

Reply 1: We gratefully appreciate for your valuable suggestion. We are sorry for the confusion caused by us that we did not explain it clearly. In fact, results in our research have been adjusted for all potentially available confounding factors, e.g., age, sex, hypertension, lipid treatment, antiplatelet treatment, NSAIDs therapy, CCI, and HCUI (full model) had been adjusted in multiple analyses. In the stratified analysis, we also adjusted for the other covariates. Compared to how important that glucosamine is to explain the incidence of CVD, we focused more on the association between glucosamine use and CVD risk. We further added a forward stepwise adjustment model analysis in our revision version and it also demonstrated the stability of our results. (line 193-194, and 344-348, Table A1)

Table A1. Stepwise adjustment model of association between glucosamine use with cardiovas-cular disease (CVD).

Adjusted covariates

HR (95%CI) for glucosamine use

None

1.11 (1.09, 1.13)

+ Age

1.06 (1.04, 1.07)

+ Sex

1.09 (1.08, 1.11)

+ Hypertension

1.10 (1.08, 1.11)

+ Lipid treatment

1.07 (1.05, 1.09)

+ Antiplatelet treatment

1.06 (1.05, 1.08)

+ NSAIDs therapy

1.06 (1.05, 1.08)

+ CCI

1.09 (1.07, 1.10)

+ HCUI (full model)

1.10 (1.08, 1.11)

NSAIDs, non-steroidal anti-inflammatory drug; CCI, Charlson Comorbidity Index; HCUI, health care utilization index; HR, hazard ratio; CI, confidence interval.

-      C2: As the authors know: the extremely high numbers of participants result in extremely low p-values (Table 1), therefore a thorough evaluation of the impact by confounding factors within one overall model would be strongly encouraged.

R2: Thank you for your advice. We agree with the reviewer that the extremely high numbers of participants would result in extremely low p-values. As mentioned in Answer 1, compared to how important that glucosamine is to explain the incidence of CVD, we focused more on the association between glucosamine use and CVD risk. Glucosamine use is not a determinant of CVD in patients with osteoarthritis, but it is positively associated with the risk of CVD, especially in adherent group. In addition, glucosamine use was still significantly associated with cardiovascular risk after multiple adjustments. In Table A1, models showed the similar results even in the overall model.

-      C3: The present reviewer appreciates the mentioning of the specific patient group of persons with osteoarthritis in many sections of the manuscript by the authors. The outcome of the present study only applies on this specific group. As presented in the Discussion section the population consists of somewhat older adults suffering from osteoarthritis. Unfortunately no physical burden of this disease is mentioned but it could well be that the pain per participant is very different which could result in the use of all kinds of treatments in those persons with the heaviest burden and having its impact on the outcome. In Table 1 it is clear that the those persons consuming glucosamine are the ones with the highest comorbidities. The question then becomes to which extent this aspect is covered by the covariates, such as CCI and HCUI. Please provide more thoughts on the impact of characteristics of the participants on the outcome.

R3: Thank you so much for your precious comments. First, as noted by the reviewer, pain was an important confounder in this study. Considering data on pain severity cannot be obtained directly from the database, we adjusted the use of NSAIDs to reduce the confounding as much as possible because NSAIDs use is associated with pain in patients with osteoarthritis. Second, although the incidence of CVD was affected by comorbidities and health care utilization, the association of glucosamine use with risk of CVD is still significant after adjusting for CCI and HCUI (Table A1). In the overall model, HR for CCI was 1.494 (1.488, 1.500), and HR for HCUI was 1.003 (1.003, 1.003), while HR for glucosamine use was1.095 (1.077, 1.113). Actually, the impact of baseline differences on outcomes is one of the key issues we tried to balance and discuss. In a real-world population of this big size, the difference of baseline characteristics did exist. However, as discussed in line 298-308, we adjusted for multiple covariates and used multiple sensitivity analyses (including PSM) to validate the results, and we think further researches are still needed to convince our results with more covariates.

-      C4: Do the authors believe that important covariates were not present in the database, such as information on genetic aspects of acquiring CVD issues?

R4: We are very grateful for your helpful comments. Genetic aspects not available in our database do influence the risk of CVD, but we suppose that there would be no correlation between genetic factors and glucosamine use - that is, belonging to a nondifferential misclassification - would make the results in this research more conservative. We have added this discussion in line 305-308.

-     C5: In Europe the combination of glucosamine with chondroitin is very often used. Could the authors comment to which extent their findings would be different between persons with and without chondroitin or is there no information in the database on this parameter?

R5: We would like to thank the reviewer for your valuable advice. The medication information of chondroitin is temporarily vacant in our study. Your comments provide a commendable direction for our future research. We have added this part into our discussion to highlight the focus of future research in line 308-310. We would like to share our future research results with this reviewer.

-      C6: In the present manuscript we are dealing with an incidence parameter and not with a prevalence parameter: the authors were looking for new cases. Would the authors believe that there might be a difference between incidence versus prevalence scoring with respect to the consumption of glucosamine on the risk of acquiring CVD issues?

R6: Thank you for your comments. The authors do believe that there might be a difference between incidence versus prevalence. First, this research is a longitudinal study in chronological order -- glucosamine exposure precedes the onset of CVD -- but not a cross-sectional study. This means that CVD must be a new case, and the parameter (outcome) must be incidence. Second, prevalence means both new cases and old cases, but in reality they are quite different. Old cases might change their habits and behaviors due to their disease, leading to reverse causation and potential confounding. Therefore, we only dealt with an incidence parameter but not a prevalence parameter in this research.

-     C7 Please provide some explanation with respect to Table 2, more specifically how is the Confidence Interval calculated? Moreover is it possible that within one person more than one incidence score (new cases after treatment of older cases) could be detected? What is the definition here.

R7: We are very grateful for your comments. We are sorry for the confusion caused by us that we did not explain it clearly. The 95% CI of incidence rate was calculated using the following formula:

95% CI = the number of events / cumulative person-years ± 1.96 * sqrt (the number of events)/ cumulative person-years

One person only had one incidence score. The incident CVD, coronary heart disease (CHD), and stroke events of subjects were observed from the index date to December 31, 2017, which were identified by the primary diagnosis at each inpatient or outpatient encounter during the follow-up. Each participant had only one outcome in this study. We have added this part in line 104-105, and 131-133.

-     C8: In that light please explain lines 80 – 81 more thoroughly: what is specifically meant here?

R8: Thank you so much for your precious advice. We are sorry for the confusion caused by us that we did not explain it clearly. A two-year wash-out period was used to identify the new patients with osteoarthritis and the new glucosamine users, which meant that participants diagnosed with osteoarthritis or taking glucosamine within two years before baseline were excluded as old cases or former users. We have added this part in line 81-83.

-     C9: Instead of the construction of cohorts (various adherence groups) for the glucosamine adherence group why not use the adherence scores as a continuous factor? The power of this approach might be substantially higher than the one selected by the authors.

R9: We gratefully appreciate for your valuable suggestion. In fact, we have tried to use the adherence score as a continuous variable, and our results are still stable after adjusting it. We included it in the stratified analysis because the risk of CVD varied among three adherence groups, and the results of stratified analysis were highly meaningful – patients who used glucosamine adherently had a 68% increase in risk for CVD compared with that in non-users. In 2002, Benner et al.[1] recommended that medications could be divided into three adherence groups according to PDC as we mentioned in methods part in line 92-101. We thought more information would be provided in the stratified analysis.

-     C10: Another aspect in the above is the application of the propensity score matching. There are however authors who suggest that this PSM is not the best way to counteract selection bias and causal estimation, such as King&Nielsen (2019), or omitted variable bias. This reviewer appreciates some more discussion on this topic, such as in the Discussion section.

R10: We are very grateful for your suggestion. This comment inspired us to use more matching methods for mutual verification in the future and made the authors realize the potential risks of PSM. King and Nielsen suggested that PSM might lead to more imbalance between groups, but from the PSM results in this research, we obtained a stable and more conservative result. This increased the reliability of our findings. We have added the citation of King&Nielsen (2019) and further discussion about PSM in line 301-303, and 454-455.

-     C11: Sensitivity analyses as performed. Please discuss to which extent the high number of persons/person years limit the applicability of these analyses.

R11: Thank you so much for your advice. Although the huge number of participants showed a relatively low difference in outcome between groups with low p-values, which limited the applicability of these analyses when we divided participants into only two groups -- glucosamine users and non-users, we still found adherent glucosamine users had a 68% increase in risk for CVD. Moreover, the authors would think that the stability of effect value is more important than the size of p value in sensitivity analysis. We have added this part into discussion in line 280-283.

C12: Minor issues:

-          Please erase lines 154 – 156

-          Grammar issues, e.g.

o   Line 18: “were selected out” versus “were selected”

o   Line 44: “were” versus “are”

o   Lines 49 – 50: “were with” versus “contained”

o   Lines 63: “it collected” versus “it contained”

o   Line 91: “take” versus “took”

o   Line 205: “was showed” versus “showed”

R12: We thank the reviewer very much for pointing the issues out. We have revised all the language issues you mentioned above.

  •  

We highly value this reviewer's suggestions, which inspires us to develop predictive models using database resources in the future. We carefully revised the manuscript according to the comments of the reviewers, and added a table to support our conclusions. Thanks again to the editors and reviewers for their comments. Please do not hesitate to contact us if you have any further questions.

References

  1. Benner, J.S.; Glynn, R.J.; Mogun, H.; Neumann, P.J.; Weinstein, M.C.; Avorn, J. Long-term persistence in use of statin therapy in elderly patients. Jama 2002, 288, 455-461, doi:10.1001/jama.288.4.455.

Reviewer 2 Report

I congratulate the authors.

Author Response

Dear Prof. Bruno Trimarco, Prof. Dr. Gaetano Santulli,

Thank you so much for your review concerning our manuscript entitled “Glucosamine use and risk of cardiovascular diseases in patients with osteoarthritis: a longitudinal study in China”. We gratefully thank all editors and reviewers for your time and useful suggestions. The comments are very precious, valuable, and helpful, which would significantly improve the quality of our manuscript. In addition, one reviewer suggested that our manuscript needs extensive English revisions, and we accept any form of English revisions. We have read the comments carefully and uploaded files of the revised manuscript. Revisions in the text are highlighted in ‘track changes’ and the responses to the comments are as follows.

#Reviewer 2

  • Comments

I congratulate the authors.

Thank you so much for your precious comments and recommendation concerning our manuscript entitled “Glucosamine use and risk of cardiovascular diseases in patients with osteoarthritis: a longitudinal study in China”. We really appreciate your comments. Revisions in the text are highlighted in ‘track changes’ and the responses to the comments are as follows. Please do not hesitate to contact us if you have any further questions.